# Preclinical Evidence of Progesterone as a New Pharmacological Strategy in Human Adrenocortical Carcinoma Cell Lines

**DOI:** 10.3390/ijms24076829

**Published:** 2023-04-06

**Authors:** Mariangela Tamburello, Andrea Abate, Elisa Rossini, Ram Manohar Basnet, Daniela Zizioli, Deborah Cosentini, Constanze Hantel, Marta Laganà, Guido Alberto Massimo Tiberio, Salvatore Grisanti, Maurizio Memo, Alfredo Berruti, Sandra Sigala

**Affiliations:** 1Section of Pharmacology, Department of Molecular and Translational Medicine, University of Brescia, 25123 Brescia, Italy; 2Section of Biotechnology, Department of Molecular and Translational Medicine, University of Brescia, 25123 Brescia, Italy; 3Oncology Unit, Department of Medical and Surgical Specialties, Radiological Sciences, and Public Health, University of Brescia and ASST Spedali Civili di Brescia, 25123 Brescia, Italy; 4Department of Endocrinology, Diabetology and Clinical Nutrition, University Hospital Zurich (USZ) and University of Zurich (UZH), 8091 Zürich, Switzerland; 5Medizinische Klinik und Poliklinik III, University Hospital Carl Gustav Carus Dresden, 01307 Dresden, Germany; 6Surgical Clinic, Department of Clinical and Experimental Sciences, University of Brescia at ASST Spedali Civili di Brescia, 25123 Brescia, Italy

**Keywords:** adrenocortical carcinoma, ACC cell lines, progesterone, zebrafish model, metastasis, invasion, migration, apoptosis, autophagy

## Abstract

Background: Adrenocortical cancer (ACC) is a rare malignancy with a dismal prognosis. The treatment includes mitotane and EDP chemotherapy (etoposide, doxorubicin, and cisplatin). However, new therapeutic approaches for advanced ACC are needed, particularly targeting the metastatic process. Here, we deepen the role of progesterone as a new potential drug for ACC, in line with its antitumoral effect in other cancers. Methods: NCI-H295R, MUC-1, and TVBF-7 cell lines were used and xenografted in zebrafish embryos. Migration and invasion were studied using transwell assays, and MMP2 activity was studied using zymography. Apoptosis and cell cycle were analyzed by flow cytometry. Results: Progesterone significantly reduced xenograft tumor area and metastases formation in embryos injected with metastatic lines, MUC-1 and TVBF-7. These results were confirmed in vitro, where the reduction of invasion was mediated, at least in part, by the decrease in MMP2 levels. Progesterone exerted a long-lasting effect in metastatic cells. Progesterone caused apoptosis in NCI-H295R and MUC-1, inducing changes in the cell-cycle distribution, while autophagy was predominantly activated in TVBF-7 cells. Conclusion: Our results give support to the role of progesterone in ACC. The involvement of its analog (megestrol acetate) in reducing ACC progression in ACC patients undergoing EDP-M therapy is now under investigation in the PESETA phase II clinical study.

## 1. Introduction

Adrenocortical carcinoma (ACC) is a rare and aggressive malignancy, with an annual incidence of 0.7–2.0 new patients per million/population [1]. While surgery represents the cornerstone of the treatment of localized ACC, advanced/metastatic disease is still hard to treat. Standard systemic therapy in this setting is based on mitotane (M) administered alone [1] or in association with the EDP regimen (etoposide, doxorubicin, and cisplatin) [2,3]. Unfortunately, the progression occurs almost invariably after less than 18 months, and there are currently no defined lines of treatment available [3,4], since neither molecular target agents nor immunotherapy has been shown to obtain substantial clinical benefit [5,6]. Therefore, new therapeutic strategies are needed, and research is currently focused on finding new drugs and improving the efficacy of existing ones [7].

Progesterone (Pg) is a lipophilic hormone that plays a fundamental role in normal developmental, reproductive functions and disease processes [8]. Despite Pg being dysregulated in different types of cancer and associated with cancer progression [9], several studies have demonstrated its antitumoral effect by regulating various cancer cell processes, including proliferation, apoptosis, angiogenesis, and autophagy, in addition to migration and invasion [10]. Indeed, Pg inhibits the proliferation of breast cancer and osteosarcoma cells [11] and contributes to the decreased progression of colorectal cancer [12,13,14], by inhibiting cell proliferation [15]. Pg also promotes apoptosis in endometrial cancer [16] and HeLa cells, arresting the progression from the G1 phase to the S phase [17] and it could induce autophagy in astrocytes [18,19,20]. Moreover, increasing data suggest that Pg inhibits migration and invasion in breast [21], ovarian [22], and endometrial cancer cells, thereby reducing their invasive potential [23].

Despite the limited knowledge about the effect of Pg in ACC, data published in the last few years by our group demonstrated the role of this hormone as an antitumoral drug in this setting. Indeed, by investigating the therapeutic use of abiraterone acetate in preclinical models of ACC, our group demonstrated that its antiproliferative effect is due to the increased production of Pg. This hormone, through its receptors (PgRs), was able to reduce cell viability in NCI-H295R cells and primary secreting ACC cultures, in a concentration-dependent manner [24], involving both genomic and nongenomic pathways [25]. In addition, we showed that the reduction of β-catenin nuclear translocation may contribute to the Pg cytotoxic effect and that Pg combined with other drugs such as mitotane [25] or the CDK4/6 inhibitor ribociclib [26] enhances their antineoplastic activity. More recently, we showed that the metastasis-derived cell models, namely MUC-1 and TVBF-7 cells (formerly ACC115m primary cells), were also targets of the Pg effect, albeit with a lower response, due to weaker PgR expression compared to NCI-H295R cells, thus strengthening the role of PgR in mediating the effect of Pg in reducing both cell proliferation and cell viability [27].

Here, we focused our interest primarily on the Pg effect on ACC metastatic cell lines MUC-1 and TVBF-7, studying whether Pg could influence ACC cell growth, invasiveness, and metastasis formation in both in vivo and in vitro models. Furthermore, we sought to shed light on the molecular mechanism underlying the cytotoxic effect of Pg in these ACC experimental models. This study aims to provide preclinical evidence for an association between Pg treatment and recurrence reduction in ACC patients.

## 2. Results

### 2.1. Pg Inhibited Tumor Growth and Metastases Formation in the Zebrafish/Tumor Xenograft Model

We first evaluated whether the cytotoxic effect elicited by Pg in vitro [25,27] was also present in vivo, taking advantage of the experimental model of ACC cells xenografted in kdrl-GFP zebrafish embryos. Preliminary experiments were conducted to evaluate the Pg toxicity on wildtype (AB) strain zebrafish embryos (Appendix A). On the basis of these results, the concentrations of 6.25 µM and 12.5 µM were chosen to evaluate the effects of Pg on ACC cell growth in the xenografted zebrafish model. Figure 1A,B show results obtained by exposing embryos to 6.25 µM Pg, demonstrating that Pg was able to significantly reduce the tumor mass in each cell line (NCI-H295R: −30.15 ± 7.22%, *p* < 0.05; MUC-1: −41.5 ± 10.47%; TVBF-7: −34.03 ± 4.28%; *p* < 0.0001). When embryos were treated with 12.5 µM, we observed a high mortality rate in embryos xenografted with the MUC-1 cell line, which did not allow to calculate the Pg effect on xenograft, while a significant reduction in tumor area was observed in the NCI-H295R (−46.48 ± 4.82%; *p* < 0.0001) and TVBF-7 (−37.53 ± 2.15%; *p* < 0.0001) cell lines (Appendix A). According to their metastatic origin, MUC-1 and TVBF-7 cells were found to be able to metastasize, albeit with some differences in terms of rate and metastases localization (Figure 2). No metastases was observed in embryos xenografted with NCI-H295R cells, in line with their origin from a primary ACC. Pg was able to reduce the rate of embryos with MUC-1 cells migrating to the caudal region from 62.5 ± 9.6% to 10.8 ± 0.85% (*p* < 0.05). TVBF-7 cells were able to form metastases at a lower rate, compared with MUC-1 cells (metastases-positive embryos: 12.07 ± 7.31%), and metastases was mostly localized in the pericardial zone. No embryos with metastases were found in the Pg-treated group. Collectively, these data indicated that Pg suppressed ACC cell growth and metastases formation in this xenografted zebrafish model.

### 2.2. Pg Suppressed the Migration and Invasion Ability of ACC Cells

To confirm and deepen the results obtained in the in vivo experimental model, we decided to further investigate the motility of ACC cells using in vitro approaches such as the transwell assay and wound healing assay. Figure 3 shows the effect of Pg on the invasion and migration ability of each ACC cell line with some representative images. Results obtained in vitro validated those obtained in vivo. MUC-1 cells displayed a high invasive capability, confirming results obtained in the zebrafish models. The invasion and migration abilities of MUC-1 cells were strongly reduced by Pg (−49.42 ± 5.34% invasive cells, *p* < 0.005; −42.76 ± 1.17% migrated cells, *p* < 0.0001; cell separation distance: 421.7 µm ± 16.64 µm in Pg-treated cells vs. 303.1 µm ± 18.61 µm in untreated cells; *p* < 0.0001). A similar antimetastatic effect of Pg was observed in TVBF-7 cells, confirming in vitro the results obtained in vivo, showing a low invasion ability (−36.85 ± 7.83% invasive cells, *p* < 0.05; −62.29 ± 16.28% migrated cells, *p* < 0.05). The transwell migration assay revealed an even lower ability to migrate compared to the ability to invade. The purely technical reason may lie in the lower incubation time used for the migration assay (22 h for migration assay and 72 h for invasion assay, as indicated in the manufacturer’s protocols). On the other hand, the wounding assay proved that TVBF-7 cells could migrate and that Pg hindered the edge reconnection (cell separation distance: 606.7 µm ± 9.14 µm in Pg-treated cells vs. 480.9 µm ± 14.13 µm in untreated cells; *p* < 0.0001).

In line with the previous observations, NCI-H295R showed a limited ability to migrate or to invade, which was not significantly modified by Pg treatment.

### 2.3. Pg Interfered with Metalloprotease 2 (MMP-2) Activity

The activity of metalloproteases (MMPs) is crucial for the cell invasion process [28]. Among these, ACC tissues specifically express high levels of MMP-2 compared to normal adrenal tissues, representing an unfavorable prognostic factor [29]. Our preliminary qRT-PCR analysis on MMP expression (Appendix A) confirmed that, on ACC cell lines, the most expressed was MMP2, even if at different levels between NCI-H295R and the metastatic cell lines (Appendix A). The effect of Pg on MMP2 expression and secretion was, thus, studied. No significant differences were observed in MMP2 gene expression, as well as in its inhibitors (TIMP1 and TIMP2), after treatment in each ACC cell line (Appendix A). Western blot analysis performed on conditioned media revealed that Pg significantly reduced the secreted MMP2 levels compared to untreated cells in metastatic cell lines (Figure 4A). Figure 4B shows the zymography results for the activity of MMP-2 equally impaired by Pg treatment (Pg-treated MUC-1 cells −27.82 ± 1.17% vs. untreated, *p* < 0.001; Pg-treated TVBF-7 cells −24.19 ± 4.43%, *p* < 0.05). No significant differences were observed in MMP-2 levels and activity after treatment on NCI-H295R cells. The low MMP2 levels observed in NCI-H295R cells compared with metastatic cell lines, are in line with their primary origin and to the lack of capability to invade both in vitro and in vivo models. These results strengthen the involvement of MMP-2 in the progression and invasiveness of ACC.

### 2.4. Pg Exerted a Long-Lasting Inhibitory Effect after Treatment

With the aim of evaluating a possible clinical application for Pg, we investigated whether Pg was able to induce in ACC cells a cytotoxic insult that, in addition to a certain percentage of cell death, could reduce cell viability in the remaining live cells. Cells were, thus, treated with the respective IC_50_ values for 4 days and thereafter kept in culture for up to 10 days in a complete, drug-free medium. Cell viability was evaluated at the end of treatment and at different times after discontinuation, as shown in Figure 5. The effect of Pg on NCI-H295R cells lasted up to 2 days after withdrawal. Cells then recovered from the cytotoxic insult, and the cell viability after 10 days of withdrawal was similar to untreated cells (Figure 5A). In the metastatic cell line MUC-1, Pg induced a cytotoxic insult that affected cell viability after drug discontinuation (Figure 5B). Indeed, cell viability significantly continued to decrease up to 10 days after drug withdrawal (−83.2 ± 1.5% compared to untreated cells; *p* < 0.0001). Figure 5C shows the effect of the drug withdrawn in the other metastatic cell line TVBF-7. In this experimental model, even if the cells seemed to slowly recover, the reduction in cell viability remained significant (−39.4 ± 2.07% compared to untreated cells; *p* < 0.0001). Taken together, these results show that, in metastatic ACC cell lines, Pg treatment induced cell damage that also progressed in the absence of the drug.

### 2.5. Pg Exerted Cytotoxic Effects on ACC Cells through Autophagy-Related Apoptosis

With the aim of identifying the possible Pg-activated intracellular pathways, with a particular interest in mechanisms underlying the cytotoxic effect in metastatic cell lines, we investigated whether, as demonstrated in the NCI-H295R cells, Pg could activate apoptosis [25]. We, thus, conducted a time course of Pg exposure (24, 48, 72, or 96 h) at its IC_50_, using annexin V/PI staining to detect apoptotic cell death induced by Pg in the experimental ACC cell models used in the present study. Figure 6 and Table 1 show that a significant increase in apoptotic cells emerged, depending on the cell model examined. In NCI-H295R cells (Figure 6A, Table 1, and Appendix A), exposure to Pg led to an increase in the percentage of apoptotic cells at each timepoint, which reached statistical significance after 48 h of treatment and was present at 72 h and 96 h, albeit with a lower effect. Mechanisms underlying this trend of response are still under investigation. In MUC-1 cells (Figure 6B, Table 1, and Appendix A), Pg treatment induced a significant decrease in viable cells and a significant increase in apoptotic cell ratio after 72 and 96 h of treatment (72 h viable cells: untreated: 96.13 ± 0.72%, Pg-treated cells: 79.25 ± 5.39%, *p* < 0.01); (96 h viable cells: untreated: 92.91 ± 1.11%, Pg-treated: 78.58 ± 0.48%, *p* < 0.0001). In MUC-1 cells, no significant increase in necrotic cells was observed. In the TVBF-7 cell line (Figure 6C, Table 1, and Appendix A), no significant differences were observed, except for an increased percentage of viable cells after 96 h of treatment (untreated viable cells: 83.69 ± 3.85%, Pg-treated viable cells: 95.90 ± 1.54%, *p* < 0.05). Thus, to elucidate the death mechanism underlying the cytotoxic effect of Pg on TVBF-7 cells, we analyzed the expression of microtubule-associated protein light chain 3 (LC3) with particular attention to LC3-II, the mature form of LC3-I, which can be used as a biomarker of autophagosome formation (Figure 7). Figure 7C shows that Pg induced autophagy on TVBF-7 cells with a significant increase in both LC3-I and LC3-II levels in a time-dependent manner. In NCI-H295R (Figure 7A) and MUC-1 (Figure 7B) cell lines, where apoptotic cell death was observed, a slight rise in LC3-I and LC3-II levels could be appreciated, albeit not significant and with a trend tending to decline. These observations confirm that, in a complex interplay between autophagy and apoptosis pathways, apoptosis is promoted when autophagy is inhibited [30], at least in MUC-1 cells. NCI-H295R cells displayed a complex interplay, which needs to be further investigated. Taken together, these results suggest that autophagy and apoptosis occur hierarchically or independently to contribute to Pg-induced ACC cell death.

### 2.6. Pg Induced Changes in the Cell-Cycle Distribution

Pg treatment also influences the cell-cycle distribution in NCI-H295R cells [25]. The cell-cycle distribution in MUC-1 and TVBF-7 ACC cell lines after Pg treatment was studied. Figure 8A,B show that, in NCI-H295R cells, treatment with Pg increased the proportion of cells in the G1 phase after both 72 h and 96 h (72 h: +10.65 ± 4.1%, 96 h: +3.08 ± 0.05%; *p* < 0.05). An increase in the G2 phase fraction was observed in MUC-1 cells at the same timepoints (72 h: +11.1 ± 0.05%, 96 h: +8.18 ± 1.42%; *p* < 0.005), along with a decrease in cells in the S phase after 96 h of treatment (−7.2 ± 1.02%; *p* < 0.005). No changes in the cell-cycle distribution were observed in Pg-treated TVBF-7 cells compared to controls. Representative DNA histograms for each cell line are shown in Appendix A.

## 3. Discussion

Identifying new molecular pathways druggable in the pharmacological armamentarium continues to be a major challenge in ACC therapy. PgRs are expressed at different intensities in both normal and neoplastic adrenal glands [27,31], and they mediate the cytotoxic effect of Pg in ACC cell models [25,27], suggesting the possibility to investigate the benefits of another pharmacological tool over the usual systemic therapy. Indeed, our previously published results indicated that Pg is effective in inducing a cytotoxic effect in several cell models of ACC, albeit at lower potency in metastatic ACC cell lines compared to the NCI-H295R cell line. This effect seems to be strictly related to the level of PgR expression since the lower potency could be due to the lower PgR expression in metastatic lines compared to H295R cells [27]. Thus, the evaluation of the PgR expression during the pathological staging could be of interest to stratify the patients in a view to personalized medicine.

As the drug treatment in advanced/metastatic ACC patients that progressed after EDP-M is still an unmet clinical need, this study was mainly dedicated to the evaluation of Pg activity in ACC metastatic cell lines (MUC-1 and TVBF-7), and to study its effect on NCI-H295R cells, which represent the widely known experimental cell model of ACC. We already demonstrated that Pg exerts a cytotoxic effect in the above-mentioned cell lines [25,27]. Here, we demonstrated that the Pg cytotoxic and antiproliferative effect was also observed in the in vivo model of zebrafish embryos xenografted with ACC cells. This result is of particular interest, as the Pg concentration that induced a significant reduction in the tumor mass area of each ACC cell line was remarkably lower than the in vitro effective concentration. In addition to being a useful tool for in vivo first drug screening [32], this animal model allowed us to demonstrate that Pg inhibits metastases formation in zebrafish embryos xenografted with metastatic cell lines such as MUC-1 and TVBF-7. As the metastatic process is associated with collective cell migration and invasion, which are common phenotypes in epithelial cancers, we deepened this aspect. First, using different in vitro assays, we demonstrated that Pg reduces migration and invasion in ACC cell lines, in line with previous reports demonstrating that Pg inhibits cell invasion and migration in other tumors [21,22]. Next, we investigated the molecular mechanism underlying the antimetastatic effect of Pg, by studying whether this effect involved MMP-2.

Indeed, the dissemination of malignant neoplasms is assumed to require the degradation of different components of the matrix and basement membrane. MMPs are responsible for the degradation of several extracellular matrix (ECM) components. There are over 20 recognized MMPs, each with specific substrate requirements and structural domains. Among these, MMP-2 and MMP-9 are highly associated with tumor dissemination and invasiveness [33]. ACC tissues specifically express MMP-2 (as also compared to normal adrenal tissue), and the elevated MMP-2 expression in ACC is an unfavorable prognostic factor [29]. Accordingly, we found that Pg impairs the secretion and the activity of MMP2, suggesting that it could mediate at least in part the effect of Pg in suppressing the invasion of metastatic ACC cells. Despite this, Pg did not influence the gene expression of MMP2 and its inhibitors TIMP1 and TIMP2, leading us to speculate that it may act at different levels on the process of MMP2 maturation and secretion. This mechanism is worth further exploration in future studies.

Drug withdrawal experiments have shown variable results in the different ACC cell models. NCI-H295R cells were able to quickly recover after cytotoxic insult and restart proliferation, while metastatic cell models, MUC-1 and TVBF-7 cells, kept a low proliferation rate, evident especially in MUC-1 cells. This in vitro result is especially relevant considering the possible clinical application of this treatment in patients with advanced ACC, in whom maintaining a cytotoxic effect with low proliferative rate cells would represent a very important aspect.

Lastly, we investigated the intracellular mechanisms inducing the cytotoxic effect of Pg in our ACC cell models. Our results demonstrated that Pg induced G1-phase arrest in the cell cycle only in the NCI-H295R cell line, but G2-phase arrest in MUC-1 cells, culminating in apoptotic cell death. On the other hand, the TVBF-7 cell line did not show any inhibition; although a cytotoxic effect was present, no increase in apoptotic or necrotic cells was observed. This discrepancy may be due to the different regulatory mechanisms of cell growth inhibition in specific ACC cell types that give support to the well-known heterogeneity of ACC [34]. The mechanism underlying the cytotoxic effect of Pg on TVBF-7 was seemingly linked to an increase in LC3-II levels, representing the activation of autophagy. The role of autophagy in the ACC context is challenging because it plays a double-edged role not only as a survival response to chemotherapeutic drugs, resulting in treatment failure, but also as an important mechanism underlying tumor cell suicide [35,36,37]. The cytoprotective function of autophagy is activated in many circumstances by suppressing apoptosis, and this evidence justifies the observation of an increase in the cell viability in TVBF-7 cells exposed to Pg. However, we would like to underline that interminable autophagy has been shown to enhance anticancer drug-induced cell death [38]. Remarkably, this study demonstrated that Pg triggered cytotoxicity due to autophagy or apoptosis depending on the ACC cell type. Accordingly, Pg is able to induce apoptosis [39,40] or autophagy [18,19,20] depending on cellular contexts; accordingly, it has already been shown that Pg induces cell-cycle modifications in cell lines derived from human tumors, with sometimes opposite results that depend on the tumor origin [41,42].

We are aware that these are preliminary results; further studies are needed to deepen the molecular features underlying this difference to shed light on the complex interplay between cell-cycle regulation and apoptosis [43].

However, with these results, we would like to give evidence of the role of Pg in inducing apoptosis or autophagy and cell-cycle modification in ACC experimental cell models.

## 4. Materials and Methods

### 4.1. Cell Lines

The human NCI-H295R cell line, derived from a primary ACC in a female patient [44], was obtained from the American Type Culture Collection (ATCC, Manassas, VA, USA) (RRID:CVCL_0458) and cultured as indicated. The MUC-1 cell line, established from a neck metastasis of an EDP-M-treated male patient, was kindly donated by Dr. Hantel and cultured as suggested [45]. Additionally, the new ACC cell line TVBF-7 [46] was established from a primary culture derived from a perirenal lymph-node metastasis of a male ACC patient who underwent progression after EDP-M. A detailed description of these three cell lines was provided by Sigala et al. [47]. All three cell lines were periodically tested for mycoplasmas and authenticated by genetic profiling using polymorphic short tandem repeat loci with the PowerPlex Fusion system (Promega, BMR Genomics Cell Profile service, Padova, Italy).

### 4.2. Cell Treatments

Cells were treated for 4 days using their respective Pg IC_50_ values (H295R: 25 µM; MUC-1: 67.58 µM; TVBF-7: 51.56 µM) [25,27]. Pg (Merk, Milan, Italy) was dissolved in DMSO in a stock solution of 100 mM, aliquoted, and stored at −20 °C. All treatments were conducted in charcoal/dextran-treated serum.

### 4.3. Measurement of Cell Apoptosis

The Pacific Blue^TM^ annexin V/SYOXTM AADVancedTM apoptosis kit (Invitrogen, Thermo-Fisher Scientific, Milan, Italy), was used to investigate Pg-induced cell death. ACC cells (5 × 10^5^ cells/well) were seeded in a six-well plate in complete medium; 24 h later, cells were treated for 24, 48, 72, or 96 h with their Pg IC_50_ values. Cells were collected, washed with ice-cold PBS, resuspended in binding buffer, and stained with Pacific Blue^TM^ annexin V/SYOXTM AADVancedTM, according to the manufacturer’s instructions. Cells were then analyzed using a MACSQuant10 Analyzer (Miltenyi Biotec GmbH, Bielefeld, Germany), using unlabeled cells as a negative control. Quantification of apoptosis was determined by FlowJo v10.6.2 software(TreeStar, Ashland, OR, USA).

### 4.4. Cell-Cycle Analysis

Flow cytometric cell-cycle analysis was performed as described [48], with minor modifications. Briefly, untreated and Pg-treated cells were fixed, treated with Rnase A (12.5 µg/mL) (Thermo-Fisher Scientific, Milan, Italy), stained with propidium iodide (40 µg/mL) (Invitrogen, Thermo-Fisher Scientific, Milan, Italy), and analyzed by flow cytometry using a MACSQuant10 Analyzer (Miltenyi Biotec GmbH, Bielefeld, Germany) for cell-cycle status. Data were analyzed using FlowJo v10.6.2 software (TreeStar, Ashland, OR, USA).

### 4.5. Drug Withdrawal Experiments

ACC cells were plated in 24-well plates and treated with their IC_50_ value of Pg for 4 days. At the end of the treatment, the drug-containing medium was replaced by a fresh complete medium without the drug, and cell viability was evaluated at different times, up to 10 days (about twice the doubling time). Cells were analyzed for cell viability using the 3-(4,5-dimethyl-2- thiazol)-2,5-diphenyl-2H-tetrazolium bromide (MTT) dye reduction assay according to the manufacturer’s protocol (Merck, Milan, Italy).

### 4.6. Fish and Embryos Maintenance

Zebrafish were maintained and used according to EU Directive 2010/63/EU for animal use following protocols approved by the local committee (OPBA) and authorized by the Ministry of Health (Authorization Number 393/2017). Adult transgenic line Tg (kdrl:EGFP) and wildtype zebrafish lines were maintained as described in [49]. Breeding of adult male and female zebrafish was carried out through natural crosses, and embryos were collected and raised in fish water with incubation at 28.5 °C until the experiments. Embryos at 24 h post fertilization (hpf) were treated with 0.003% 1-phenyl-2-thiourea (PTU) to prevent pigmentation. After the conclusion of the experiments, the zebrafish embryos were euthanized with 400 mg/L tricaine (ethyl 3-aminobenzoate methanesulfonate salt; Sigma-Aldrich, Milan, Italy)

### 4.7. Tumor Xenograft

Tumor xenograft experiments were performed as described in [50] with minor modifications. Briefly, to evaluate the toxic effect of Pg on the zebrafish model, 48 hpf wildtype embryos (AB) were divided into different groups as indicated and maintained in PTU/fish water, to which solvent (DMSO) or increasing concentrations of Pg (10, 25, 50, and 100 µM) were added. After 3 days (T3), effects of Pg were observed. To evaluate the effect of Pg on tumor growth, Tg (kdrl:EGFP) zebrafish embryos at 48 hpf were dechorionated, anesthetized with 0.042 mg/mL tricaine, and microinjected with the labeled tumor NCI-H295R, MUC-1, and TVBF-7 cells (CellTrackerTM CM-Dil Dye, Thermo Fisher Scientific, Milan, Italy) into the subperidermal space of the yolk sac. Microinjections were performed with a FemtoJet electronic microinjector coupled to an InjectMan N12 manipulator (Eppendorf Italia, Milan, Italy). Approximately 250 cells/4 nL were injected into each embryo (about 25 embryos/group); embryos were maintained in PTU/fish water in a 32 °C incubator to allow tumor cell growth. Pictures of injected embryos were acquired using a Zeiss Axiozoom V13 (Zeiss, Jena, Germany) fluorescence microscope, equipped with Zen 2.3 Blu software, 2 h after cell injection (T0). Then, 6.25 µM Pg, 12.5 µM Pg, or solvent (DMSO)was directly added to the PTU/fish water in Pg-treated and untreated experimental groups. After 3 days (T3), the effects of the drug on cancer growth were scored by taking pictures, as previously described, to measure the tumor areas of each group at T0 and T3 using Zen 2.3 Black software (ZEISS, Jena, Germany). Embryos with metastases (i.e., the presence of at least one fluorescence dot outside the site of injection) were counted, and some representative xenografted embryos were fixed, embedded in low melting agarose, and imaged using an LSM 510 confocal laser microscope equipped with an Achropla 10×/0.25 objective.

### 4.8. Migration and Invasion Assay

The motility and invasive capabilities of untreated and Pg-treated ACC cells were explored using transwell assays. For the migration assay, a QCMTM Chemotaxis 24-well cell migration assay with an 8 µm pore size (Merk, Milan, Italy) was used. The in vitro invasiveness of cells was evaluated using an ECMatrix cell invasion assay with an 8 µm pore size (Merck, Milan, Italy). Untreated or treated cells were added into the upper chamber of migration or invasion inserts according to the manufacturers’ protocols. The chemoattractant gradient was created using a medium enriched with 10% fetal bovine serum. After incubation in a humidified tissue culture incubator for 22 h (for migration assay) or 72 h (for invasion assay), the cells in the interior of the chamber that did not penetrate the membrane were cleaned up. Penetrated cells were stained; then, after washing, inserts were air-dried. Images were acquired using an Olympus IX51 optical microscope (Olympus, Segrate, Italy) equipped with a 10× objective. Subsequently, staining was eluted, and absorbance was detected using an EnSight Multimode Plate Reader (PerkinElmer, Milan, Italy) at 560 nm.

### 4.9. Wound Healing Assay

To evaluate the wound healing assay, cells were cultured in a six-well plate until reaching 90–100% confluency, at which point a scratch was created using a p200 pipette tip. The medium was removed, and cells were washed in PBS before adding the medium conditioned with vehicle or Pg. Five different areas along the scratches of each well were analyzed by optical microscopy after 0 and 24 h following the induced damage. The distance between each edge of the scratch was monitored under an Olympus IX51 microscope and was measured using the software NIH ImageJ Software V1.52a.

### 4.10. Western Blot

Cells were homogenized in cold RIPA buffer, and total protein concentrations were determined using the Bio-Rad Protein Assay (Bio-Rad Laboratories, Segrate, Italy). Equal amounts of proteins (30 µg) were run in 10% polyacrylamide gels and transferred onto a polyvinylidene fluoride (PVDF) membrane. Anti-LC3 primary antibody (Sigma-Aldrich, Milan, Italy Cat# L8918, RRID: AB_1079382) and secondary HRP-labeled anti-rabbit antibody (Promega, Milan, Italy Cat# W4011, RRID: AB_430833) were used. Densitometric analysis of the bands was performed using NIH ImageJ Software V1.52a. To evaluate secreted MMP2 levels, the conditioned medium was obtained as described in [51]. Briefly, equal amounts of untreated and Pg-treated cells were seeded in serum-free medium (1 × 10^6^ cells/mL medium). After 24 h, the conditioned medium was collected, stored at −80 °C for at least 24 h, and subsequently freeze-dried. The residues were resuspended in the same volume of PBS. Next, proteins were separated by electrophoresis on a 4–12% NuPAGEbis-tris gel system (Life Technologies, Monza, Italy), electroblotted to a PVDF membrane, and finally detected using an anti-MMP2 primary antibody (Proteintech, Planegg-Martinsried, Germany Cat# 10373-2-AP, RRID: AB_2250823).

### 4.11. Zymography

Novex 10% Zymogram Plus (Gelatin) gels (Thermo Fisher Scientific, Milan, Italy) were used to measure secreted MMP2 activity in a conditioned medium, obtained as described in Section 4.10. Different aliquots of the same conditioned medium samples were used for both Western blot and zymography. The proteases were run under denaturing conditions and visualized as clear bands against a dark background using a renaturing, developing, and staining protocol (Thermo Fisher Scientific, Milan, Italy).

### 4.12. Gene Expression

Preliminary experiments to evaluate the gene expression of different metalloproteases (MMPs) in NCI-H295R, MUC-1, and TVBF-7 cell lines were performed using the Human Tumor Metastasis RT2 Profiler PCR Array (Qiagen, Milan, Italy). RNA isolation, treatment with DNase, and RNA purification were performed using the Qiagen kit (Qiagen, Milan, Italy). For cDNA synthesis, the RT2 First-Strand Kit (Qiagen, Milan, Italy) was used. PCR was performed with VIIA7 (Applied Biosystems, Monza, Italy) using RT2 SYBR Green qPCR Mastermix (Qiagen, Milan, Italy) as the fluorochrome.

For the other gene expression analysis, total RNA was extracted from cells using the RNeasy kit (Qiagen, Milan, Italy), and 1 µg was transcribed into cDNA, using murine leukemia virus reverse transcriptase (Promega, Milan, Italy). Gene expression was evaluated by q-RT-PCR (ViiA7, Applied Biosystems, Monza, Italy) using SYBR Green as the fluorochrome. Specific primers for MMP2, TIMP1, and TIMP2 were as follows: MMP2 (F: 5′–ACGACCGCGACAAGAAGTAT–3′ and R: 5′–ATTTGTTGCCCAGGAAAGTG–3′), TIMP1 (F: 5′–GGGACACCAGAAGTCAACCA–3′ and R: 5′–GGCTTGGAACCCTTTATACATC–3′), and TIMP2 (F: 5′–AAGCGGTCAGTGAGAAGGAA–3′ and R: 5′–TCTCAGGCCCTTTGAACATC–3′). Expression levels were normalized to the β-actin mRNA level of each sample, obtained from parallel assays.

### 4.13. Statistical Analysis

Statistical analysis was carried out using GraphPad Prism software (version 5.02, GraphPad Software, La Jolla, CA, USA). One-way ANOVA with Bonferroni’s correction was used for multiple comparisons. Where appropriate, the unpaired *t*-test was used. Unless otherwise specified, data were expressed as the mean ± standard error of the mean (SEM) of at least three experiments run in triplicate. A *p*-value < 0.05 was considered statistically significant.

## 5. Conclusions

In conclusion, our results further strengthen the evidence of a possible therapeutic role of Pg in ACC, supporting the hypothesis of a positive role of Pg treatment in ACC patients’ care. It is worth underlining that the Pg analog megestrol acetate is already part of cancer supportive care, thus providing another pharmacological tool with a well-known risk/benefit profile compared to usual systemic therapy in reducing ACC progression in patients undergoing EDP-M therapy. This hypothesis is currently under study in the ongoing double-blind, placebo-controlled randomized phase II clinical trial PESETA (EudraCT Number: 2020-004530-38).

## Figures and Tables

**Figure 1 ijms-24-06829-f001:**
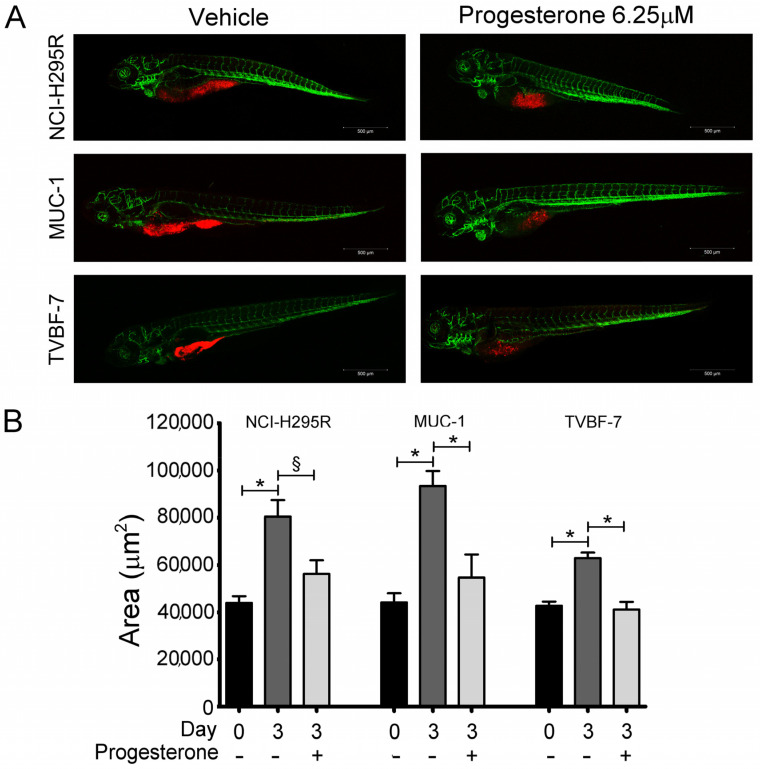
Pg induced a reduction in the tumor xenograft area of ACC cells. (**A**) Representative, lateral-view pictures of Tg (kdrl: EGFP) control and Pg-treated embryos at 120 hpf. Each ACC cell line was labeled with a red fluorescent lipophilic dye while the embryo endothelium was labeled with a green fluorescent protein reporter driven by the kdrl promoter. Images were acquired using a Zeiss LSM 510 META confocal laser scanning microscope at 10× magnification. (**B**) Tumor areas at 48 hpf (T0—the start of treatment) and 120 hpf (T3—end of treatment) of drug-treated and vehicle-treated groups measured using Zen 2.3 Black software from ZEISS. Data are shown as the mean of independent experiments ± SEM. * *p* < 0.0001; § *p* < 0.05.

**Figure 2 ijms-24-06829-f002:**
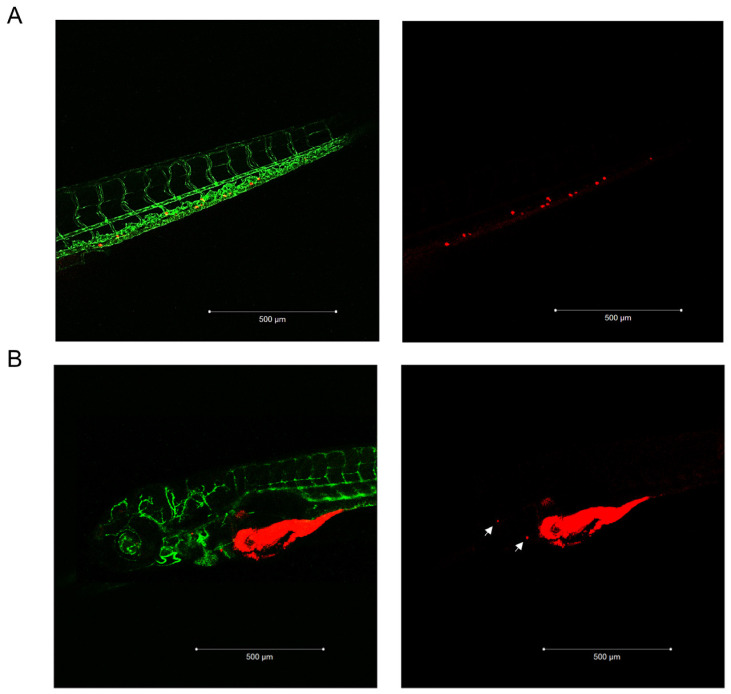
MUC-1 and TVBF-7 ACC cells induced metastases formation in zebrafish embryos. (**A**) Enlargement of the embryo tail with metastasized MUC-1 cells. (**B**) A representative acquisition of the metastases of TVBF-7 cells in the pericardial area of the embryo is shown. Cells are labeled with a red fluorescent lipophilic dye while the embryos’ endothelium is labeled with a green fluorescent protein reporter driven by the kdrl promoter. Images were acquired at 120 hpf using a Zeiss LSM 510 META confocal laser scanning microscope at 10× magnification.

**Figure 3 ijms-24-06829-f003:**
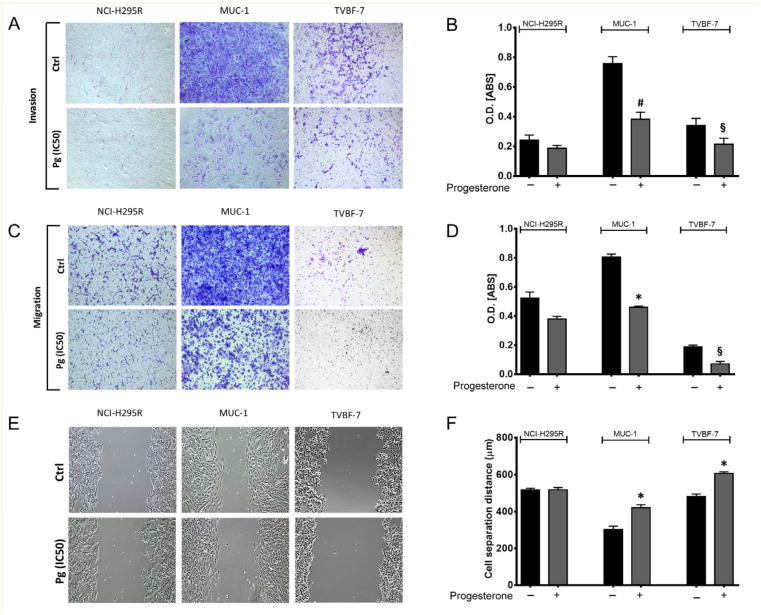
Pg suppressed ACC cell invasion (**A**) and migration (**C**) ability in transwell assays. Images were acquired using an Olympus IX51 optical microscope at 10× magnification. (**B**,**D**) Quantification of the number of invasive and migrated cells was analyzed using the absorbance of the staining detected at 560 nm. (**E**) Representative images of wound healing assay used to detect migrated ACC cells. (**F**) The distance between each edge of the scratch was measured using NIH ImageJ Software V 1.52a. Images were acquired using an Olympus IX51 optical microscope equipped with a 10× objective. Data are shown as the mean ± SEM of three independent experiments. * *p* < 0.0001, # *p* < 0.001, § *p* < 0.05 vs. untreated cells.

**Figure 4 ijms-24-06829-f004:**
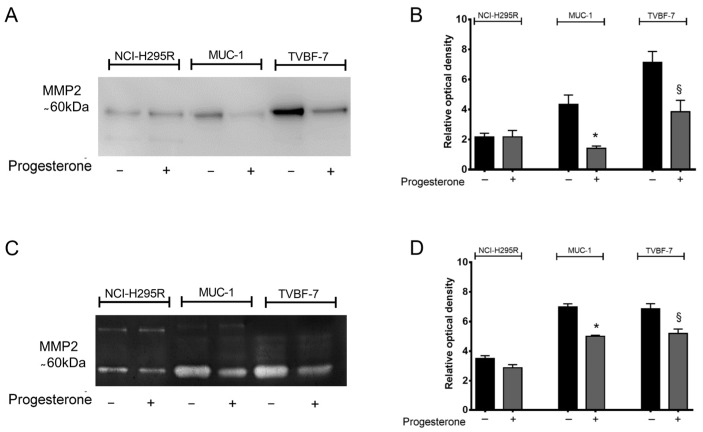
The secretion and the activity of MMP 2 were decreased after Pg treatment. (**A**) Representative Western blot of MMP2 in the conditioned medium secreted by ACC cells. (**C**) Representative zymogram of MMP2 in the conditioned medium secreted by ACC cells. (**B**,**D**) Quantification of Western blots and zymography. Results are presented as the mean relative optical density ± SEM of three independent experiments. * *p* < 0.0001, § *p* < 0.05 vs. untreated cells.

**Figure 5 ijms-24-06829-f005:**
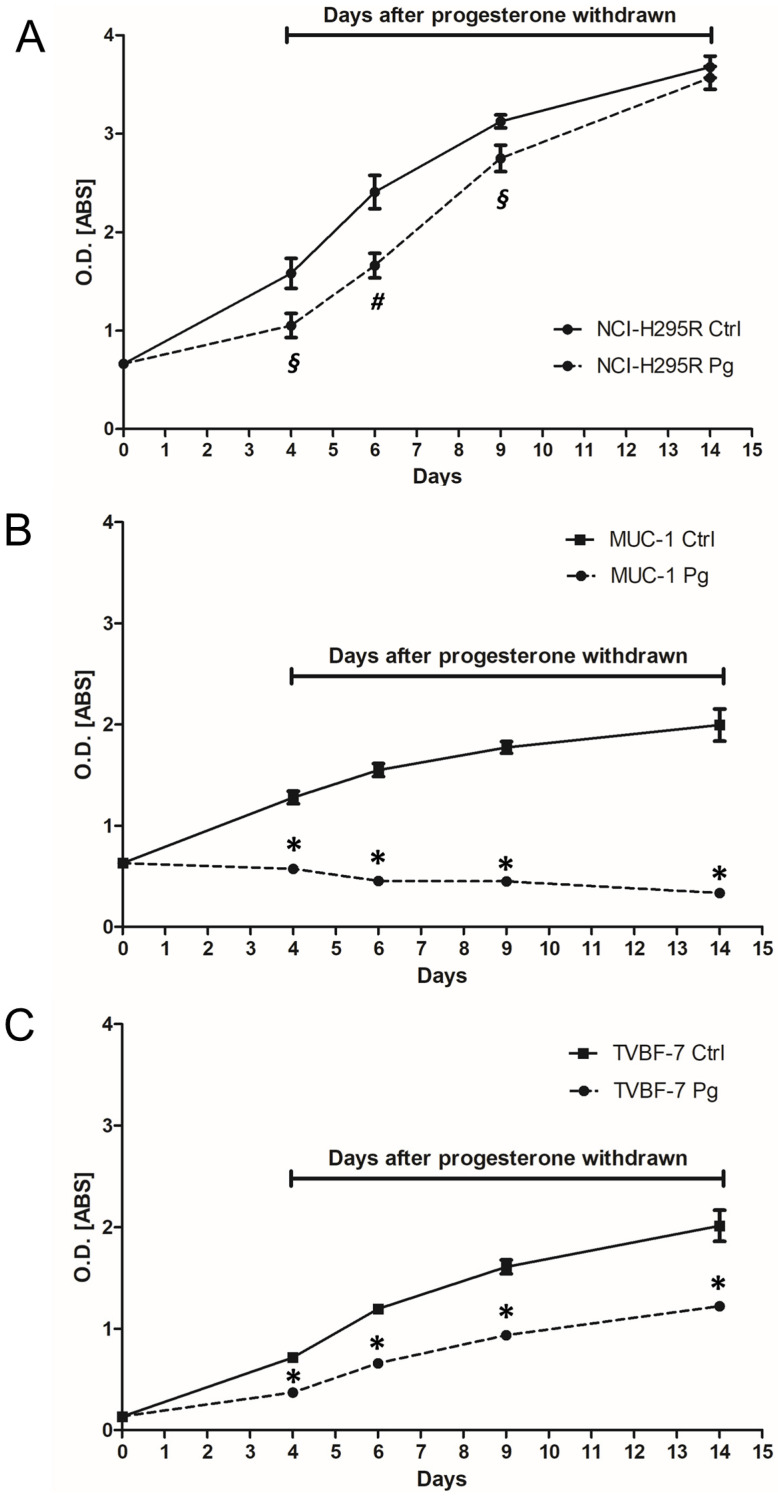
Effect of drug withdrawal on NCI-H295R (**A**), MUC-1 (**B**), and TVBF-7 (**C**) ACC cell lines. Cells were treated for 4 days with respective IC_50_ values of Pg; then, the drug was withdrawn from the medium, and cells were kept in culture for up to 10 days. The cell viability time course was measured by the MTT assay. Points represent the mean ± SEM of at least three experiments performed in triplicate. * *p* < 0.0001, # *p* < 0.001, § *p* < 0.05 vs. untreated cells.

**Figure 6 ijms-24-06829-f006:**
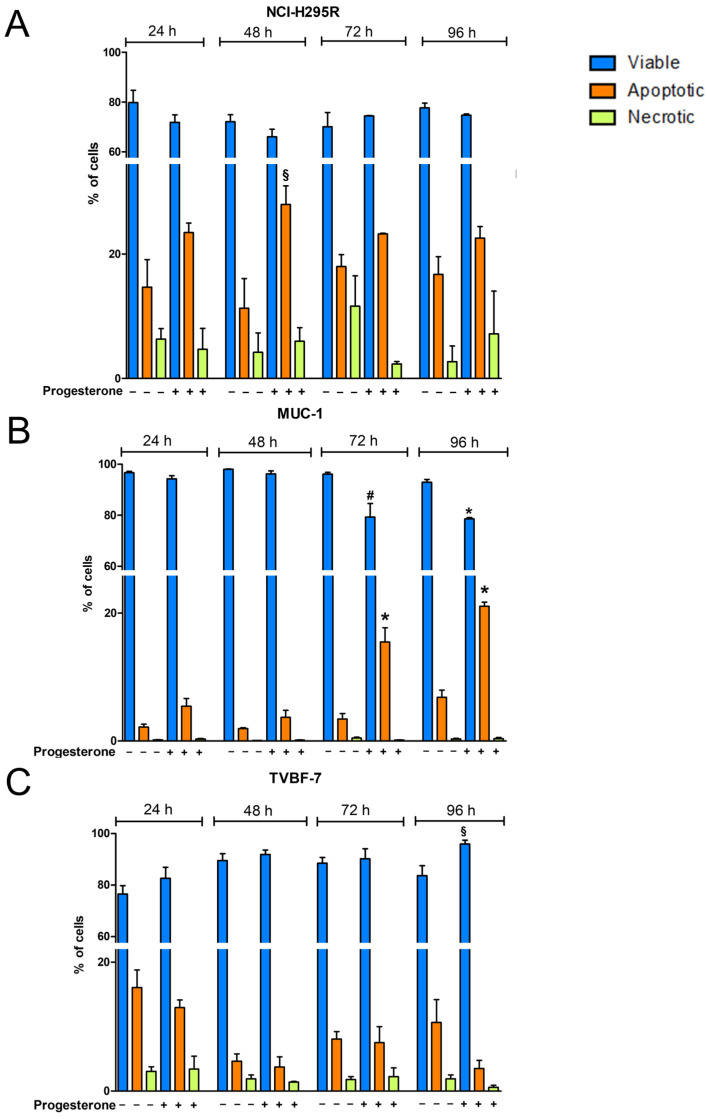
Pg promoted apoptotic cell death in NCI-H295R (**A**) and MUC-1 (**B**) cell lines but not in TVBF-7 cells (**C**). Cells were treated for 24, 48, 72, or 96 h using their Pg IC_50_ values, stained with Pacific Blue^TM^ annexin V, and analyzed by flow cytometry. Histograms representative of the mean ± SEM of three experiments are shown. * *p* < 0.0001, # *p* < 0.001 cells, § *p* < 0.05 vs. untreated cells.

**Figure 7 ijms-24-06829-f007:**
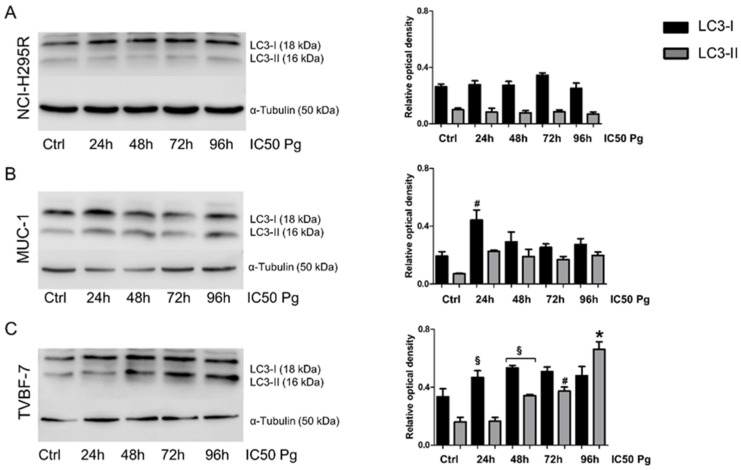
Pg triggered autophagy cell death in TVBF-7 cell lines. Representative Western blot of LC3-I and LC3-II protein levels in NCI-H295R (**A**), MUC-1 (**B**), and TVBF-7 (**C**) are shown. On the right side, quantification results are presented as a relative optical density means ± SEM of three independent experiments. * *p* < 0.0001, # *p* < 0.001, § *p* < 0.05 vs. untreated cells.

**Figure 8 ijms-24-06829-f008:**
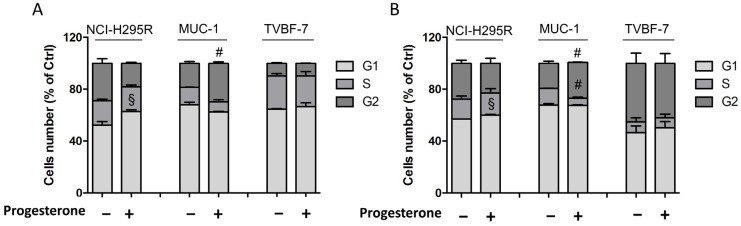
Pg-induced changes in the cell-cycle distribution of ACC cell lines. Cells were treated for 72 h (**A**) and 96 h (**B**) with respective IC_50_ values of Pg, stained with propidium iodide, and analyzed for DNA content by flow cytometry. Histograms representative of the mean ± SEM of three experiments are shown. # *p* < 0.001, § *p* < 0.05 vs. untreated cells.

**Table 1 ijms-24-06829-t001:** Percentage of apoptotic cells observed in the Pg treatment time course in ACC cell lines.

	NCI-H295R		MUC-1		TVBF-7	
Pg	−	+	−	+	−	+
24 h	14.66 ± 4.44	23.45 ± 1.52	2.16 ± 0.45	5.43 ± 1.19	16.07 ± 2.73	12.95 ± 1.18
48 h	11.28 ± 4.76	27.95 ± 3.02 §	1.90 ± 0.14	3.68 ± 1.09	4.63 ± 1.14	3.73 ± 1.58
72 h	17.98 ± 1.93	23.23 ± 0.13	3.42 ± 0.86	15.47 ± 2.21 *	8.07 ± 1.15	7.50 ± 2.48
96 h	16.72 ± 2.84	22.57 ± 1.83	6.81 ± 1.13	21.08 ± 0.64 *	10.64 ± 3.55	3.51 ± 1.24

§ *p* < 0.05 vs. untreated cells; * *p* < 0.001 vs. untreated cells.

## Data Availability

The datasets generated and analyzed during the current study are available from the corresponding author upon reasonable request.

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
