# Peer review of "Preclinical Evidence of Progesterone as a New Pharmacological Strategy in Human Adrenocortical Carcinoma Cell Lines"

_ijms, 2023, doi:10.3390/ijms24076829_

Round 1

Reviewer 1 Report

The manuscript titled "Progesterone as a potential drug for adrenocortical cancer: in vitro and in vivo evidence" is an informative and well-written study. The authors investigate the potential use of progesterone as a therapeutic agent for adrenocortical cancer (ACC), a rare malignancy with a poor prognosis. The authors tried to explore the role of progesterone in ACC treatment, based on its anti-tumoral effects in other cancers. The study uses NCI-H295R, MUC-1, and TVBF-7 cell lines, which were also xenografted in zebrafish embryos. The authors found that progesterone had a long-lasting effect in metastatic cell lines, MUC-1 and TVBF-7, and induced apoptosis in NCI-H295R and MUC-1 cells. Furthermore, progesterone reduced tumor area and metastasis formation in embryos injected with metastatic cells. The study has several strengths, including the use of both in vitro and in vivo models, the investigation of multiple cell lines, and the exploration of potential mechanisms of action. The results provide evidence to support the role of progesterone as a potential therapeutic agent for ACC, however I have following concerns/comments/ suggestions.

Abstract: The abstract is well-written and provides an overview of the research work.

Introduction: The introduction is well-written and provides sufficient background information to introduce the topic of the study.

In Figure 1 and Table 1 authors showed Percentage of apoptotic cells observed in the Pg-treatment time-course in ACC cell lines, I find these results to be interesting, however I have some concerns regarding the apoptosis percentage observed in the NCI-H295R, MUC-1, and TVBF-7 cell lines. Specifically, the authors observed a higher apoptosis percentage in NCI-H295R cells at 48 hours, which appeared to decrease at 72 and 96 hours. The authors should provide an explanation for this variation in the results.

In the MUC-1 cell line, the apoptosis percentage trend seems to decrease at 48 hours compared to 24 hours, and then increased significantly at 72 and 96 hours. Similarly, in the TVBF-7 cell line, the apoptosis percentage in the control group at 24 and 48 hours seems to have obvious variation, which raises concerns about potential experimental handling problems. The authors should address this possibility and provide additional information to support the reliability of their results.  I also suggest authors to provide representative Flow cytometry images in supplementary or main figure along with their analytical (bar graphs) results.  

Since their follow up results are based on these results, I also suggest Authors to see the expression of apoptosis/necrosis markers along with autophagy markers to conclude if these phenomena occur hierarchically or independently to contribute to Pg-induced ACC cell death.

Also I am just wondering if authors have seen the effect of pg treatment in any normal cell line as well along with ACC cells???

In results 2.2, authors used 72 and 96 hours’ time point for their cell cycle analysis specially in  NCI-H295R cell line, however the significantly results were observed at 48 hours. I am just wondering the reason of selecting these two time point in NCI-H295R as they did not observe any significant results based on table 1 ? Also authors should see the expression levels of cell cycle markers and provide the representative images of flow cytometery in supplementary or main figures. I am just wondering if cells were synchronized before cell cycle analysis.

Results 2.3. The conclusion of the results is missing.

2.4 Authors mentioned that Preliminary experiments were conducted to evaluate the Pg toxicity on wild-type (AB) strain zebrafish embryos. If possible, authors should provide these results as supplementary results.

Overall, the manuscript has the potential to contribute to the literature on ACC treatment with progesterone. However, it is recommended that the authors address the concerns mentioned above to improve the quality and reliability of their findings.

Reviewer 2 Report

The manuscript “Preclinical evidence of progesterone as a new pharmacological strategy in human adrenocortical carcinoma cell lines” by Tamburello M et al, aims at characterizing the antitumour effect of progesterone (Pg) on adrenocortical carcinoma (ACC) using an array of in vitro and zebrafish models.

The experiments reported in this paper were performed using three different ACC cell line, who have been shown to bear specific genomic alterations.

This is a well written paper showing new data on the effect of Pg on ACC cancer cells.

The authors found an effect of Pg in inducing apoptosis in H295R and MUC-1 cells, while TVBF-7 cells were unresponsive., and this phenomenon was inversely correlated to the expression of autophagy markers.

Pg also affected cell viability when removed from the medium, suggesting a long-lasting effect.

The effect of Pg was even more pronounced in zebrafish in reducing tumour mass (H295R and TVBF-7) upon transplantation, as well as migration to the caudal region (TVBF-7 and, to a less extent, MUC-1).

The effect of Pg is quite noticeable in the zebrafish model, where transplanted cells are able to self-organized in a three-dimensional fashion, more closely resembling the architecture of the tumour of origin, despite the absence of tumour associate cells, such as fibroblasts and immune cells. 

-       It would be interesting to assess differential expression of key players that directly (Pg receptor/ NR3C3) or more downstream (MMP2) in 2D vs 3D cultures.

Pg has been showed to induce the expression of its receptor in human breast, uterine, and ovarian cancer cell lines  (i.e. DOI: 10.1016/j.steroids.2016.09.004), does Pg have any effect on PgR expression in the three ACC cell lines?

Line 109: “To elucidate the death mechanism underlying the cytotoxic effect of Pg on TVBF-7 cells”, aren’t TVBF-7 cells not responders for Pg in terms of apoptosis?

Please provide information related to how results were normalised for Figure 8 when conditioned medium is used; if Pg has an effect on cell viability, fewer cells would be expected to secrete less MMP2? Meaning, is it that lower MMP2 is due to fewer cells but overall able to secrete comparable MMP2 to untreated cells?

Round 2

Reviewer 1 Report

Authors have adequately addressed all of my concerns and made the necessary revisions to meet the standards of our publication.

Reviewer 2 Report

Thank  you for taking the time to address my comments.The revisions you have made and the additional experiments you performed have improved the overall quality of your paper as well as the clarity and coherence of the study.

I believe this work makes a valuable contribution to the field and that the manuscript will be of great interest to readers gravitating in the adrenal research.